# Transgressive Behavior in Dutch Youth Sport

Nicolette Schipper-van Veldhoven [1,2,3,4,*], Joris Mulder [5], Boukje Cuelenaere [5], Robbert Zandvliet [6], Kimberly Farzan [6] and Miriam Reijnen [7]

1    Research Centre Human Movement and Education, Windesheim University of Applied Sciences, 8000 GB Zwolle, The Netherlands
2    Faculty of Behavioural, Management and Social Sciences, Mathematics and Computer Science, University of Twente, 7522 NB Enschede, The Netherlands
3    Faculty of Electrical Engineering, Mathematics and Computer Science, University of Twente, 7522 NB Enschede, The Netherlands
4    Netherlands Olympic Committee and Netherlands Sports Confederation (NOC*NSF), 6816 VD Arnhem, The Netherlands
5    Centerdata, Tilburg University, 5037 AB Tilburg, The Netherlands
6    I&O Research, Enschede/Amsterdam, 1019 GM Amsterdam, The Netherlands
7    Dutch Centre for Sports and Safety, Netherlands Olympic Committee and Netherlands Sports Confederation (NOC*NSF), 6816 VD Arnhem, The Netherlands
*    Correspondence: n.schippervanveldhoven@windesheim.nl

**Abstract:** The current article reports on the second large-scale prevalence study on transgressive behavior in sport in the Netherlands, and is a follow up of an earlier, comparable prevalence study in 2015. Using a dedicated and customized online questionnaire, approximately 4000 adults who met the inclusion criteria (18 to 50 years old and have played sports in an organized context during childhood in the Netherlands) were surveyed with respect to their experiences of childhood psychological, physical, and sexual transgressive behavior while playing sports. The survey showed that 71.7% experienced some form of transgressive behavior as a child, in which 48.6% of these events also made an impact (in other words, was significant at the time it took place). The degree of impact the event made is also related to the severity of the event. Severe emotional transgression events occurred in 22% of the youth athletes, severe physical assault events in 12.7%, and severe sexual assault events occurred in 6.9% of the youth athletes. Disabled athletes, and those competing at national and international levels, report more experiences of transgressive behavior in sport. The results are consistent with former research and indicate the need for structural attention to create a safe sports climate.

**Keywords:** prevalence; child abuse; harassment; unwanted behavior; transgressive behavior; athlete welfare; organized sport

## 1. Introduction

Sport is popular in the Netherlands. Approximately 1.9 million young people (aged between 5 and 18 years) practice a sport, of which 1.3 million participate in organized sport at a club level (NOC*NSF 2020). Sport is known as the third pedagogical environment (next to home and school) and is seen as an environment in which youngsters can develop, learn, and have fun. Researchers agree sport can lead to positive outcomes and improved social, psychological, and/or moral character (Bailey 2005; Light 2010). Popular beliefs even tend to emphasize the positive consequences of sport (Imbrogno et al. 2021). Unfortunately, sport can also lead to negative outcomes such as depression, exhaustion, eating disorders, anxiety, and somatization (Hughes and Leavey 2012; McDuff 2012; Vertommen et al. 2018) as a result of transgressive behavior within sport. Transgressive behavior is expressed as antisocial behavior, aggression, bullying, (sexual) harassment, selfishness, arrogance, non-contact, hazing, cyber, and negligence (e.g., Kavussanu 2019; Mountjoy et al. 2016). Transgressive

behavior can be described as psychological, physical, and sexual interpersonal violence (Vertommen et al. 2016). The obligation to protect athletes from harassment and abuse is embedded in the statutory governing documents of sport, including the Olympic Charter (IOC 2020a) and the IOC Code of Ethics (IOC 2020b). All athletes have a right to engage in 'safe sport'.

In the Netherlands, attention has been paid to undesired behavior in sport since the 1980s (Breedveld and van der Poel 2015). NOC*NSF has executed a preventive and sanctioning policy with regard to sexual intimidation in sport since the late 1990s (Moget and Weber 2007; Schipper-van Veldhoven 2013; Schipper-van Veldhoven et al. 2015). Flanking research was of a primarily descriptive nature or aimed at evaluations, monitoring, and good practices (e.g., Lucassen et al. 2012; Hulsebos et al. 2015; Schipper-van Veldhoven and Steenbergen 2015). The first large-scale Dutch prevalence study was only conducted in 2015, and revealed dramatic scores on transgressive behavior (Vertommen et al. 2016).

The revelation of the abuse scandal in English football at the end of 2016 led to renewed political attention towards transgressive behavior in the Netherlands. An independent commission was given the task of providing insight into the scope and nature of the problem, and give recommendations to NOC*NSF, sports federations, and sports clubs on how to improve the sports climate (De Vries et al. 2017). One of the recommendations was to periodically survey the prevalence of (sexual) harassment and abuse in sport (ibid., p. 144). Following the earlier prevalence study by Vertommen et al. (2016), the current study is therefore the second large-scale and updated prevalence study into transgressive behavior among young athletes in the Netherlands.

The main objective of the present article is to assess the prevalence of retrospectively self-reported experienced transgressive behavior in organized youth sport in the Netherlands (using a modified version of the questionnaire used in the first study), while also considering differences in gender, age, disability, sport type, sport level, relationship to the offender, and type of transgressive behavior.

## 2. Materials and Methods

### 2.1. Definitions

For our study, we adopted the definition of transgressive behavior as documented in the Dutch Sports Disciplinary Law Blueprint SI 2018 (CVSN 2018): *"Any form of unwelcome verbal, non-verbal or physical conduct that has the purpose or effect of violating the dignity of the person, in particular when an threatening, hostile, insulting, humiliating or offensive situation is created"*. We define organized sport as every recreational or competitive sporting activity that is voluntary, takes place within the context of a club or organization outside the school curriculum, and involves an element of training or instruction by an adult. This includes sport camps and organized extracurricular sporting activities at school, but excludes physical education (PE lessons) and informal physical activities (e.g., street soccer games, walking the dog, and gardening) (see Vertommen et al. 2016, p. 226).

### 2.2. Online Questionnaire

The questionnaire used to collect the data for this research is a modified version of the interpersonal violence against children in sport questionnaire used in the first prevalence study in the Netherlands (Vertommen et al. 2016). Due to advancing insights and the experiences from the previous study, the questionnaire has been drawn up differently. We have maintained the subdivisions of psychological, physical, and sexual interpersonal violence, each measured by a set of relevant items. The order and wording of some items was modified slightly, and some (contemporary) items about social media were added. Furthermore, an updated classification method of transgressive severity was added. Another important difference is the method of measuring the prevalence of transgressive behavior. Vertommen et al. (2016) used the 'low-threshold' method for this, where 'once' is the minimum to be counted for prevalence. For example, if someone indicates that they have been bullied once, this counts as prevalence. In the current study, the 'low-threshold'

is based on an event that is reported to have occurred 'once in a while'[1] (instead of once), regardless of whether the event made an impact on the respondent or not. The reason this was adjusted is because the pre-test showed that respondents had difficulty remembering whether some events happened once or a few times. It was therefore decided to ask respondents whether an event occurred 'once in a while' or 'regularly'. This categorization provides a clearer distinction in reporting the frequency of an event.

The questionnaire was drawn up according to a funnel structure. At the beginning of the questionnaire, a framework was outlined in which abuse or transgressive behavior could have taken place (demographic and descriptive section). Outlining a framework gives respondents context to reconstruct events which minimizes the risk of memory effects as much as possible (Hardt and Rutter 2004). After the framework, questions regarding their experiences of transgressive behavior were asked step by step, ranging from intimidation to (sexual) violence. The questionnaire started with a number of selection questions. The further the respondents completed the questionnaire, the more in-depth they went into possible transgressive behavior that they experienced (psychologically, physically, or sexually) during sport in childhood. The resultant questionnaire consisted of 15 items on psychological transgressive behavior, including being bullied, humiliated, threatened, and ignored; 11 items on physical transgressive behavior, such as being pushed, shaken, beat up, and forced overtraining; and 21 items on sexual transgressive behavior, including sexual harassment and abuse, and distribution of (nude) photos or videos.

*2.3. Definition versus Experience*

The definition of transgressive behavior, as documented in the Dutch Sports Disciplinary Law Blueprint SI 2018, was used to determine the total events defined as transgressive behavior (TB). In addition to the total number of reported events, an extra dimension was added which asked about the respondent's experience of each reported event. Although the total number of reported events were used to determine the actual prevalence of TB, we felt the need to offer respondents the chance to add a level of nuance to each of the events. For instance, when a trainer yells at a soccer player during a training session, this officially counts as TB, but for one player the impact of this event can be more serious than for another player. Therefore, in addition to reporting whether an event happened (once) in a while or regularly, respondents also reported whether or not each particular event had made an impact on them. This way, we aim to draw a complete picture of the reported events based on (1) the frequency of the event (once in a while or regularly), and (2) how the respondent experienced the event (did or did it not leave an impact on them).

The first online version of this 'new' questionnaire was pre-tested by means of in-depth cognitive interviews. During this form of interviewing, 10 test respondents completed the questionnaires in the presence of a researcher and gave feedback on content, duration, and lay out. Where necessary, the questionnaire was adapted based on the results of the cognitive interviews. The second online version was piloted in the target population to test the adjustments from the qualitative pre-test and to determine whether the questionnaire was technically satisfactory. The trial included 300 panel members, of which 106 panel members completed the questionnaire. The conclusion of this quantitative test was that no further adjustments were necessary.

*2.4. Severity Classification*

Some events make more of an impact than others. Therefore, in the questionnaire, respondents were first asked which events they experienced, and if it was possible to distinguish whether this occurred occasionally or regularly. Subsequently, they were asked whether or not the experiences had made an impact.

A severity score index was constructed based on the severity scores used in the research of Vertommen et al. (2016). A small expert group judged each of the 47 items from 1 to 3 (low, medium, high), indicating how they perceived the severity of the reported incident(s) in relation to the frequency of the event, and whether the event had made an

impact on the respondent. It had 3 levels, namely mild (occasionally, no impact, including mild severity), moderate (regularly, no impact, including medium severity) and severe (including all high-severity events) (Table 1).

**Table 1.** Response classification based on expert-rated event severity, and respondent-rated event frequency and impact.

|  | Frequency | | |
|---|---|---|---|
|  | Once (in a while), no impact | Regularly, no impact | Once or regularly, made an impact |
| Event severity 0.5 (low) | **Mild** | **Mild** | **Moderate** |
|  | e.g., someone yelled at you | e.g., you were being bullied | e.g., gossip or lies were spread about you |
| Event severity 1 (medium) | **Mild** | **Moderate** | **Severe** |
|  | e.g., you were the subject of sexist jokes | e.g., someone pushed you | e.g., WhatsApp messages containing unwanted sexual content |
| Event severity 2 (high) | **Severe** | **Severe** | **Severe** |
|  | e.g., someone hit you with an object | e.g., being forcibly restrained | e.g., you were forced to have penetrative sex (oral, vaginal, or anal) |

*2.5. Procedure*

The prevalence of childhood sport-related transgressive behavior was measured using a retrospective web-based survey of adult respondents, aged between 18 and 50 years, who had engaged in organized sport as a child (up to the age of 18). Sampling and data collection were performed by Centerdata and I&O Research, research companies in the Netherlands both drawing on a longitudinal panel (respectively LISS panel[2] and I&O Research panel). For this study, drawing from both panels, a representative sample of Dutch adults (up to 50 years of age) was randomly selected. Before starting the questionnaires, respondents contained information on the content of the web survey and a hyperlink to the actual transgressive behavior questionnaire. Respondents could only proceed after agreeing with the online informed consent request.

Respondents were given options not to answer and to skip questions, and were also given the opportunity to explain answers and make remarks. After completing the questionnaire, respondents were also advised of various support services in the case they were in need of psychological or mental help.

*2.6. Recruitment, Ethics, and Consent*

During the recruitment of households for the LISS panel, respondents who agreed to participate in the panel received a confirmation email and a letter with a login code. Using the login code, they could confirm their willingness to participate and immediately start their participation in the panel. Respondents were asked to read and agree to the LISS 'informed consent'. They could confirm their consent by ticking a checkbox. Only when respondents agreed to the informed consent could they participate in the questionnaires. This confirmation and informed consent procedure, following the consent to participate given to the interviewer, ensured the double consent of each respondent to become a panel member and participate in the monthly panel questionnaires. Panel members were aware that they could terminate their panel membership at any time. It was noted that medical ethics approvals for questionnaire research among adults are not required in the Netherlands. In general, Centerdata and I&O Research abide by the European General Data Protection Regulation (GDPR).

*2.7. The Sample*

The net response of the I&O Research panel was 42% (N = 3111) and the net response of the LISS panel was 75% (N = 1981). In total, N = 5092 panel members completed the questionnaire within 4 weeks. Of these, 4166 respondents (81%) did meet the criteria for this

study (18 to 50 years old and have played sports in an organized context during childhood (younger than 18)), of which 3959 respondents fully completed the questionnaire. These 3959 respondents were the research group.

*2.8. Statistical Procedures*

The primary aim of this study was measuring the prevalence of transgressive behavior in youth sport. To be able to determine its prevalence in the Netherlands, a large enough random sample from the population was needed, as described above (see sample). Auxiliary variables have been constructed to determine the severity and frequency classification (Table 1). Descriptive statistics were used to report the sociodemographic and sports participation characteristics (Table 2) and to determine the proportions and overlap of the 3 types of reported transgressive behavior events (Table 3). To determine significant differences ($p < 0.05$) between the subgroups of gender, current age, ethnicity, participation in disabled sports, and highest sporting achievements (Table 4), Chi-square tests with Bonferroni corrections, proportions, and standardized residuals were applied. All statistical analyses were performed using IBM SPSS version 25 (Armonk, NY, USA).

**Table 2.** Sociodemographic and sports participation characteristics for all respondents.

| | | **N** | **%** |
|---|---|---|---|
| Gender | Female | 1950 | 50.7 |
| | Male | 2009 | 49.3 |
| Age | 18–24 years | 874 | 22.1 |
| | 25–34 years | 1276 | 32.2 |
| | 35–44 years | 1062 | 26.8 |
| | 45–50 years | 747 | 18.9 |
| Ethnicity | Dutch | 3160 | 79.8 |
| | Ethnic minority W * | 202 | 5.1 |
| | Ethnic minority NW ** | 191 | 4.8 |
| | Unknown | 405 | 10.2 |
| Education | Low | 352 | 8.9 |
| | Moderate | 1451 | 36.7 |
| | High | 2152 | 54.4 |
| Participation in sport for disabled children | Exclusively | 14 | 0.4 |
| | Not Exclusively | 46 | 1.2 |
| | No | 3888 | 98.2 |
| | I do not want to say | 11 | 0.3 |
| Sport organization | Recreational | 900 | 22.7 |
| | Local | 1,291 | 32.6 |
| | Regional | 1321 | 33.4 |
| | National | 374 | 9.4 |
| | International | 73 | 1.8 |
| Total | | **3959** | |

* Ethnic minority W = respondent's birth country, or that of their parents, is western, but not the Netherlands.
** Ethnic minority NW = respondent's birth country, or that of their parents, is non-western, and not the Netherlands.

**Table 3.** Overview of the frequency of self-reported TB experienced during participation in organized youth sport.

| Type of Transgressive Behavior (TB) | At Least One Reported TB Experience | | | |
|---|---|---|---|---|
| | with and without Impact | | only with Impact | |
| | N | % | N | % |
| Psychological | 2683 | 67.8 | 132 | 46.3 |
| Physical | 937 | 23.7 | 410 | 10.4 |
| Sexual | 614 | 15.5 | 285 | 7.2 |
| All three types of TB | 247 | 6.2 | 83 | 2.1 |
| At least one type of TB | 2837 | 71.7 | 1924 | 48.6 |
| No TB | 1122 | 28.3 | 2035 | 51.4 |

**Table 4.** Transgressive behavior classified according to gender, current age, ethnicity, participation in disabled sports, and highest sporting achievement.

| | | N | Psychological TB | Physical TB | Sexual TB |
|---|---|---|---|---|---|
| | | | % with and without Impact/% only with Impact | | |
| Gender | Female | 1950 | 63.3 $_a$/47.9 $_a$ | 18.5 $_a$/9.3 $_a$ | 20.2 $_a$/10.5 $_a$ |
| | Male | 2009 | 72.1 $_b$/44.7 $_b$ | 28.7 $_b$/11.4 $_b$ | 11.0 $_b$/4.0 $_b$ |
| Age | 18–24 years | 874 | 73.3 $_a$/48.3 $_a$ | 29.9 $_a$/13.0 $_a$ | 18.1 $_a$/7.6 $_a$ |
| | 25–34 years | 1276 | 70.6 $_a$/49.5 $_a$ | 24.8 $_{a,b}$/10.1 $_{a,b}$ | 15.3 $_a$/7.7 $_a$ |
| | 35–44 years | 1062 | 64.8 $_b$/44.7 $_{a,b}$ | 19.9 $_c$/8.7 $_b$ | 14.8 $_a$/5.8 $_a$ |
| | 45–50 years | 747 | 60.6 $_b$/40.80 $_b$ | 19.9 $_{b,c}$/9.9 $_{a,b}$ | 14.1 $_a$/7.9 $_a$ |
| Ethnicity | Dutch | 3160 | 72.8 $_a$/41.6 $_a$ | 23.1 $_a$/10.2 $_a$ | 14.7 $_a$/9.4 $_a$ |
| | Ethnic minority W * | 202 | 67.8 $_a$/47.3 $_a$ | 26.2 $_a$/12.3 $_a$ | 15.3 $_a$/7.1 $_a$ |
| | Ethnic minority NW ** | 191 | 67.8 $_a$/41.6 $_a$ | 30.9 $_a$/11.5 $_a$ | 19.3 $_a$/8.4 $_a$ |
| | Unknown | 405 | 65.4 $_a$/43 $_a$ | 23.0/10.4 $_a$ | 15.6 $_a$/6.7 $_a$ |
| Sports for disabled | Exclusively | 14 | 78.6 $_a$/71.4 $_a$ | 28.6 $_a$/28.6 $_a$ | 46.2 $_a$/35.7 $_a$ |
| | Not Exclusively | 46 | 75.6 $_a$/43.5 $_{a,b}$ | 35.6 $_a$/20.0 $_a$ | 26.1 $_{a,b}$/6.5 $_b$ |
| | No | 3888 | 67.8 $_a$/46.3 $_b$ | 23.6 $_a$/10.2 $_a$ | 15.3 $_b$/7.0 $_b$ |
| | Unknown | 11 | | | |
| Sport achievement | Recreational | 900 | 53.6 $_a$/36.9 $_a$ | 16.8 $_a$/7.8 $_a$ | 11.3 $_a$/5.6 $_a$ |
| | Local | 1291 | 68.1 $_b$/45.5 $_b$ | 22.9 $_b$/9.5 $_{a,b}$ | 15.6 $_b$/7.2 $_a$ |
| | Regional | 1321 | 74 $_c$/51.4 $_c$ | 25.5 $_b$/10.1 $_{a,b}$ | 16.5 $_b$/7.6 $_a$ |
| | National | 374 | 76.5 $_c$/51.2 $_{b,c}$ | 32.9 $_c$/18.8 $_c$ | 21.7 $_b$/8.8 $_a$ |
| | International | 73 | 80.8 $_{b,c}$/58.9 $_{b,c}$ | 42.5 $_c$/19.2 $_{b,c}$ | 16.4 $_{a,b}$/12.3 $_a$ |
| Total | | 3959 | 67.8/46.3 | 23.7/10.4 | 15.5/7.2 |

a, b, c: Within each demographic (e.g., gender, age, sport achievement), percentages that do not share a subscript (compared vertically) are significantly different. For instance, women reported significantly less psychological TB (63.3%) than men did (72.1%) but reported significantly more sexual TB (20.2%) than men did (11%), since the subscripts 'a and b' are different between women and men. Significance was considered $p < 0.05$ level with the Bonferroni correction. * Ethnic minority W = respondent's birth country, or that of their parents, is western, but not the Netherlands. ** Ethnic minority NW = respondent's birth country, or that of their parents, is non-western, and not the Netherlands.

## 3. Results

### 3.1. Sociodemographic and Sport Characteristics of the Study Sample

The sociodemographic characteristics of the respondents in the final sample for analysis are summarized in Table 2. In the Netherlands, around 90 percent of all Dutch adults up to the age of 50 participated in organized sport as a child (<18 years old). The respondents were evenly divided between male and female and evenly distributed between 18 and 50 years. Regarding their level of education, 9% had received lower secondary education, 37% had completed moderate secondary/vocational education, and 54% had completed higher education (obtained a degree BA, MA/MSc, PhD). The highest performance level achieved varied widely: a quarter of all respondents had played recreational sport only, two thirds played at least one of the listed sports at a competitive level, 32% played at a local level, and 34% played at a regional level. A small group of respondents played at an elite level: at a national level (10%) and/or international level (2%). Only 2% had participated in organized sport activities for disabled children, although the majority had not done so exclusively.

*3.2. Reported Prevalence of Transgressive Behavior (TB): Nature and Frequency*

Table 3 gives an overview of the frequency of self-reported transgressive behavior (TB) during the sport histories of respondents. In total, 72% of the respondents who participated in organized sport in their youth said that they experienced at least one of the three types of TB while playing sports as a child (low-threshold measure, i.e., at least one reported experience). A total of 68% of the respondents reported at least one psychological event, 24% reported one physical event, and 16% reported one sexual event during childhood sports. A total of 6% of the respondents indicated that they experienced all three forms of TB.

Not all experiences made an impact. Of those events that made an impact (in other words, were significant at the time it took place), 46% of the respondents reported experiencing at least one psychological event, 10% reported at least one physical event, and 7% reported at least one sexual event that made an impact. A total of 2% of the respondents experienced all three forms of TB that also made an impact.

The different forms of TB are shown schematically in colored circles (Figure 1), where the size of the circle is related to the extent to which this form of TB occurs. Where circles overlap, it means that there have been multiple forms of TB.

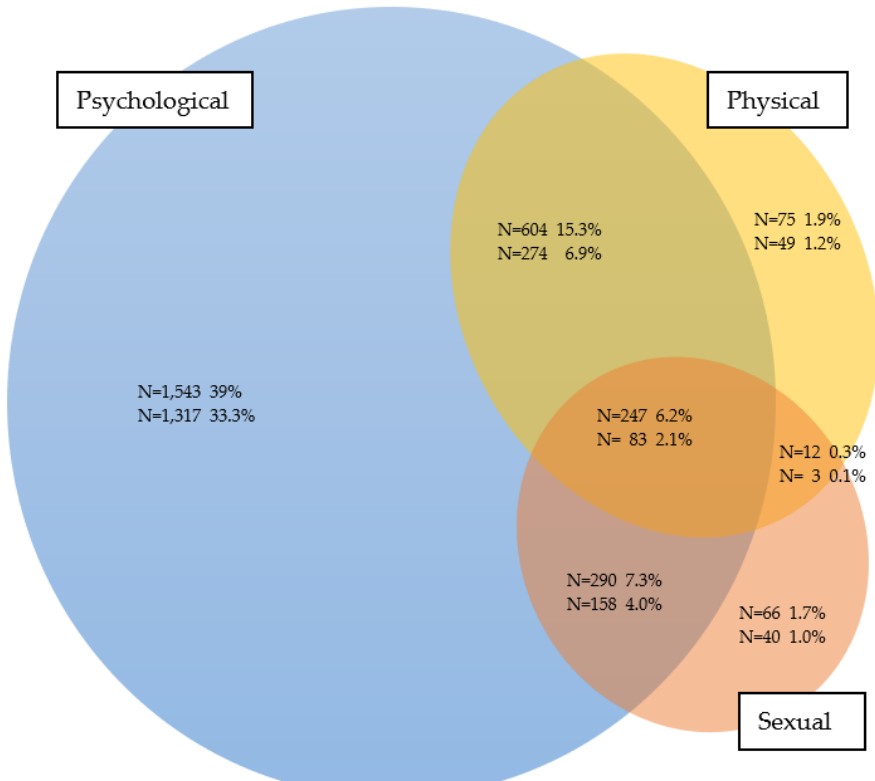

**Figure 1.** Proportions and overlap of the three types of TB, with and without leaving an impact (low-threshold measure, i.e., at least one reported experience).

The set diagram in Figure 1 shows the proportions and relations for the various types of TB. The upper frequencies and percentages represent the total reported experiences (with and without leaving an impact on the respondent). The lower frequencies and percentages represent only the reported experiences that left an impact on the respondent. Of the respondents reporting physical or sexual TB, the majority also reported at least one incident of psychological TB.

A total of 28% of respondents who played sports in an organized context in their youth indicated that they never experienced any incidence of TB, and 51% of respondents indicated that they did not experience any event of TB that made an impact.

### 3.3. Reported Prevalence of Transgressive Behavior: Differences in Gender, Age, Ethnicity, and Highest Sport Achievement

Table 4 shows that respondents who were confronted with psychological and/or physical TB during childhood organized sport participation were more often boys (72.1% and 28.7%, respectively) than girls (63.3% and 18.5%, respectively). Regarding the events that made an impact, the percentage of events that made an impact for both boys and girls was about equal. If a psychological and/or physical event happened to a boy, it seems to have made relatively less impact. In the case of sexual TB, girls (20.2% total, 10.5% with impact) report more incidents than boys (11% total, 4% with impact) in total, and only involved events that made an impact.

Experiences with sexual TB did not differ among the four age groups; respondents having participated in sports in the 1980s, 1990s, 2000s, and 2010s reported similar rates for these TB types (see Table 4, sexual TB). The number of reports of psychological TB are, however, significantly lower for the older respondent groups (64.8% and 60.1%, respectively) compared to the younger respondent groups (73.3% and 70.6%, respectively). The youngest age group (18–24) reported significantly higher physical TB (29.9%), especially compared to the older age groups.

Respondents who practiced sports at a national or international level also reported more incidents than those who played sports at a lower level. More specifically, for psychological and physical TB, the differences are significant between levels of sport achievement. For instance, concerning physical TB, recreational (16.8%) reports below the average of 23.7% and differs significantly from local (22.9%) and regional (25.5%). National (32.9%) and international (42.5%) report far above the average of 23.7%, which is significantly more than recreational, local, and regional reports. Overall, looking at the highest level the respondents achieved in their sports careers, we can conclude that there is an increased risk of TB when the level rises. Those who performed at a recreational level report the fewest TB events, whereas respondents at the international level report the most.

There is no significant difference according to ethnic background for any of the forms of transgressive behavior.

### 3.4. Severity Classification

As shown in Table 5, there seems to be an interaction effect for severity score in psychological TB when looking at gender. In the mild category, men (41.1%) have a significantly higher percentage than women (23.1%), compared to women (26.7%) versus men (17.3%) in the severe category. Among the athletes with a higher sporting achievement, the reported events were also more often severe. Concerning physical TB at the national sports achievement level, significantly more severe (21.4%) events were reported than mild (9.6%) events.

### 3.5. Relationship to the Offender

Figure 2 shows that the perpetrators were more often teammates than coaches. This applies strongly to psychological (60.2%) and physical TB (58.8%). Furthermore, other athletes (from another team) or someone else they knew were perpetrators of all three forms of TB.

**Table 5.** Expert-rated severity of the sport-related childhood transgressive behavior (TB) classified according to gender, current age, ethnicity, participation in disabled sports, and highest sporting achievement.

| | | | Psychological TB | | | Physical TB | | | Sexual TB | | |
|---|---|---|---|---|---|---|---|---|---|---|---|
| | | N | Mild | Moderate | Severe | Mild | Moderate | Severe | Mild | Moderate | Severe |
| | | | % | % | % | % | % | % | % | % | % |
| Gender | Female | 1950 | 23.1 a | 13.5 b | 26.7 c | 7.2 a | 0.8 a | 10.5 a | 10.3 a | 1.3 a | 8.5 a |
| | Male | 2009 | 41.1 a | 13.6 b | 17.3 c | 12.4 a | 1.5 a | 14.9 a | 5.1 a | 0.5 a | 5.4 a |
| Age | 18–24 years | 874 | 38.8 a | 13.7 a,b | 20.8 b | 12.4 a | 1.1 a | 16.4 a | 9.8 a | 0.8 a | 7.4 a |
| | 25–34 years | 1276 | 32.3 a, | 15.0 b | 23.3 a,b | 10.6 a,b | 1.6 b | 12.5 a | 7.2 a | 0.9 a | 7.1 a |
| | 35–44 years | 1062 | 30.1 a | 13.3 a | 21.4 a | 8.0 a | 0.8 a | 11.1 a | 8.0 a | 0.9 a | 5.8 a |
| | 45–50 years | 747 | 27.7 a | 11.1 a | 21.8 a | 8.0 a | 0.8 a | 11.1 a | 5.4 a | 1.1 a,b | 7.6 b |
| Ethnicity | Dutch | 3160 | 31.2 a | 13.9 a,b | 22.7 b | 10.0 a | 1.1 a,b | 12 b | 7.9 a | 1 a | 6.4 a |
| | Ethnic minority W * | 202 | 37.1 a | 9.9 a | 20.8 a | 8.9 a | 1.0 a | 16.7 a | 10.4 a | 0 a | 8.9 a |
| | Ethnic minority NW ** | 191 | 41.6 a | 11.1 a | 20.0 a | 12.6 a | 2.6 a | 15.7 a | 3.1 a | 1.6 b | 9.9 b |
| | Unknown | 405 | 33.8 a | 13.6 a | 18.0 a | 7.2 a | 1.5 a,b | 14.3 b | 6.4 a | 0.2 a | 8.9 a |
| Sports for disabled | Exclusively | 14 | 21.4 a | 50.0 b | 7.1 a | 0.0 a | 0.0 a | 28.6 a | 23.1 a | 0 a | 23.1 a |
| | Not Exclusively | 46 | 46.7 a | 13.3 a | 15.6 a | 6.7 a | 0.0 a,b | 28.9 b | 15.2 a | 0 a | 10.9 a |
| | No | 3888 | 32.2 a | 13.4 a | 22.2 a | 9.9 a | 1.2 a,b | 12.5 b | 7.6 a | 0.9 a | 6.8 a |
| Sport achievement | Recreational | 900 | 22.8 a | 12.1 b | 18.7 b | 6.4 a | 0.1 b | 10.3 a | 6 a | 0.8 a | 4.6 a |
| | Local | 1291 | 32.3 a,b | 12.3 b | 23.5 a | 11.3 a | 1.1 a,b | 10.5 b | 7.4 a | 0.9 a | 7.3 a |
| | Regional | 1321 | 35.9 a | 15.3 a | 22.8 a | 10.7 a | 1.8 b | 13 a | 8.2 a | 1 a | 7.3 a |
| | National | 374 | 39.8 a | 14.4 a | 22.2 a | 9.6 a | 1.9 a,b | 21.4 b | 10.7 a | 1.1 a | 9.9 a |
| | International | 73 | 44.4 a | 16.7 a | 19.4 a | 11.1 a | 0.0 a | 30.6 a | 6.9 a | 1.4 a | 6.9 a |
| Total | | 3959 | | | | | | | | | |

a, b: Within each subgroup of a demographic (e.g., women within gender, or 18–24 years within age), percentages of mild, moderate, and severe that do not share a subscript (compared horizontally) are significantly different. For instance, women reported significantly less moderate psychological TB (13.5%) than mild (23.1%) and severe (26.7%). * Ethnic minority W = respondent's birth country, or that of their parents, is western, but not the Netherlands. ** Ethnic minority NW = respondent's birth country, or that of their parents, is non-western, and not the Netherlands.

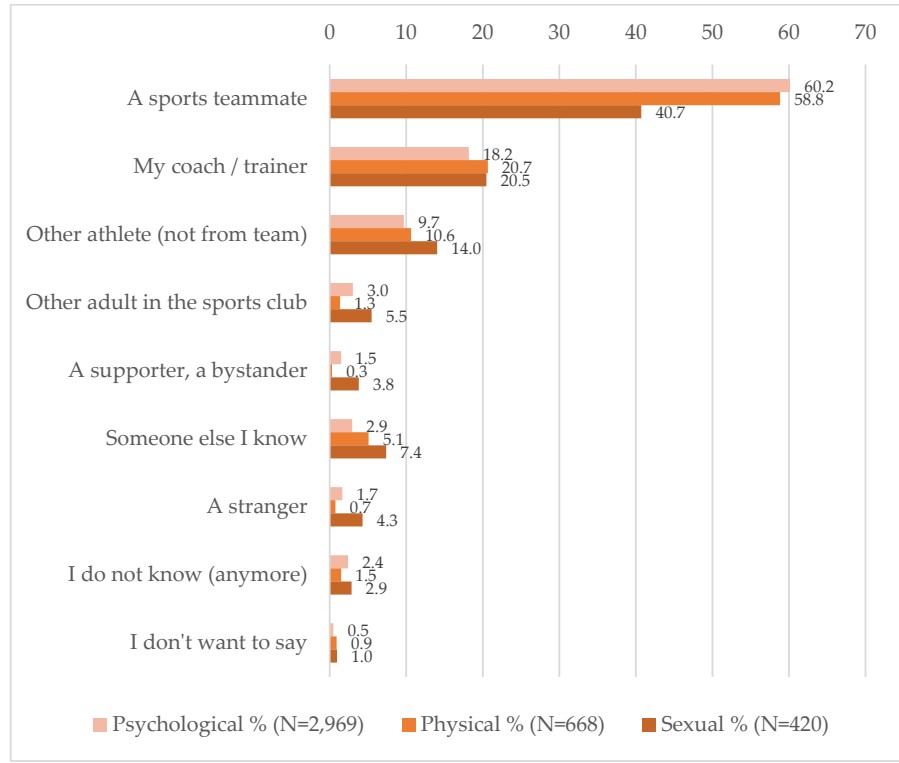

**Figure 2.** Relationship to the offender.

## 4. Discussion

The present retrospective study was designed to assess the prevalence of transgressive behavior in Dutch youth sport, using a representative sample of Dutch adults (up to 50 years of age). In this study, three forms of transgressive behavior were distinguished: psychological, physical, and sexual. An extensive series of events was surveyed for each form, ranging from being teased as a form of psychological transgressive behavior to forced penetration as a form of sexually transgressive behavior. Respondents were asked per event whether they experienced this, how often it happened (once in a while or regularly), and whether it made an impact. Our research shows that, in total, 71.7% experienced at least one event of transgressive behavior (low-threshold) and 48.6% experienced an event that also made an impact.

Psychological TB had the highest rates of occurrence, at 67.8% and 46.3%, respectively (only with impact). Physical TB occurred in 23.7% and 10.4% of cases, respectively. Sexual TB occurred in 15.5% and 7.2% of cases, respectively.

The degree of impact made by an event is also related to the seriousness of the event. On the basis of frequency and whether or not an event made an impact, supplemented with the judgment of experts, a severity score index was constructed with three levels: mild, moderate and severe. Severe psychological transgression was experienced by 22% of the young athletes, severe physical TB by 12.7%, and severe sexual TB by 6.9% of the young athletes.

Although this is the second large-scale prevalence study into transgressive behavior among young athletes in the Netherlands, and is a follow up of the first study (Vertommen et al. 2016), it can be seen as a new zero measurement. Due to advancing insight and the experiences from this earlier study, the questionnaire has been drawn up differently, in such a way that the results of the two studies cannot be compared one-on-one. Comparing our data with the first study, we see that the prevalence rates are similar for psychological TB which has the highest rates (in the first study 38%). In the second study physical TB occurred more often than sexual TB compared to the first study (physical violence 11% and sexual violence 14%). Comparing our data with the study of Alexander et al. (2011) in the United Kingdom, we see similar prevalence estimates of psychological harm (75%) and physical violence (24%). Due to differences in definitions, methods, and questionnaires, comparison with other studies is difficult (see Mergaert et al. 2016).

Our findings on gender differences show a more detailed insight than the first study. More men than women experienced psychological and physical TB, but women are more often involved in a psychological event that made an impact. We do not see this difference in physical TB. When it comes to sexual TB, girls have experienced this more often. Girls are twice as likely to experience sexual TB, they are also more likely to experience something that makes an impact, and what they experience is more serious.

Similar to the first study, respondents who are younger report more psychological TB and those who practiced sports at a national or international level were more vulnerable to all forms of TB. The risk factor of ethnicity was not confirmed in the second study.

The perpetrators are more often teammates/athletes than coaches or bystanders in all forms of TB. This is also in line with our first study (Vertommen et al. 2016).

Based on the different decades in which the respondents participated in youth sport, with even higher rates of TB in our second study, this research confirms that TB is a structural problem within organized sport. Within the Netherlands, there is a long tradition of policy making for sexual harassment in sport and a safe sport climate (Schipper-van Veldhoven et al. 2015). This policy was intensified in 2017 as the 'MeToo' movement in sport led to renewed political attention towards transgressive behavior in sport in the Netherlands (De Vries et al. 2017). In line with our data, they concluded that there is an urgent need to become more active in combating (sexual) harassment and abuse in sport. Internationally, there is an urgent call towards sport authorities to develop and implement clear and effective remedies for all types of non-accidental violence against athletes (Mountjoy et al. 2016). In this respect, McMahon et al. (2022) state that attention

should be paid, not only to sexual abuse, but especially to psychological and physical behavior that transgresses boundaries.

### 4.1. Limitations

There are limitations to retrospective self-reporting using an online panel. With respect to retrospective self-reporting, experiences often occurred years (decades) ago, which means that cognitive decline and memory effects may disrupt memory. It is also possible that respondents do not make a direct connection between their participation in sport and the experience of transgressive behavior. As a result, there is a chance that memory effects will occur, or 'recall bias' (Hardt and Rutter 2004; Baldwin et al. 2019). These effects may be enhanced by the potentially sensitive subject matter of this study. In order to minimize the influence of these disruptive effects, the measuring instrument was developed according to a funnel structure and the process was approached iteratively. It therefore had different phases of testing, feedback, and adjustment.

The sample of respondents were panel members who chose to fill out a questionnaire relatively rapidly. For this study, the composition of the dataset was checked for significant differences between sample and population response. A weighting factor was constructed to correct the deviations to the population numbers. The results in this study are presented based on the weighted figures.

Taking these limitations into account, we believe that the questionnaire and sample used in this paper provide an accurate picture of the prevalence of transgressive behavior in the Netherlands.

### 4.2. Future Research and Recommendations

In this article, we have presented the main outcomes of our survey, which produced an enormous volume of data on the different forms of transgressive behavior. Additional in-depth analyses on circumstances and impact, sport disciplines, and the alleged perpetrators of transgressive behavior will also be reported in future articles.

Negative experiences and events during youth sports can, to a greater or lesser extent, lead to persistent problems into adulthood. This can manifest itself in post-traumatic stress (PTS) symptoms, limitations in self-reliance and a reduced quality of life (e.g., Briere and Elliott 2003; Maniglio 2009). Therefore, our respondents were also asked whether they had experienced or recognized specific symptoms or life experiences. This will be reported in future articles.

Based on our research data, we demonstrated that transgressive behavior in sport is a structural problem. Therefore, we place a strong emphasis on developing interventions aimed at countering psychological and physical transgressive behavior, to broaden the horizon of safeguarding in sport to diminish not only sexual, but all forms of transgressive behavior.

**Author Contributions:** Conceptualization, N.S.-v.V., J.M., B.C. and M.R.; methodology, N.S.-v.V., J.M., B.C., R.Z., K.F. and M.R.; validation, N.S.-v.V., J.M., B.C., R.Z., K.F. and M.R; formal analysis, J.M. and B.C.; investigation, J.M., B.C., R.Z. and K.F.; resources, N.S.-v.V., J.M., B.C., R.Z., K.F. and M.R.; data curation, J.M., B.C., R.Z. and K.F.; writing—original draft preparation, N.S.-v.V. and J.M.; writing—review and editing, B.C., R.Z., K.F. and M.R.; visualization, N.S.-v.V. and J.M.; supervision, N.S.-v.V., B.C. and M.R.; project administration, B.C. and R.Z.; funding acquisition, M.R. All authors have read and agreed to the published version of the manuscript.

**Funding:** This research was funded by the Ministry of Health, Wellbeing, and Sports, Netherlands, 2019.

**Data Availability Statement:** The Dutch written report 'Grensoverschrijdend gedrag in de sport' is available at https://nocnsf.nl/media/3754/prevalentie-grens-overschrijdend-gedrag-in-de-nederlandse-sport_definitief_v20.pdf (accessed on 27 April 2022).

**Acknowledgments:** We thank Maartje Elshout from Centerdata for thoroughly programming, advising, and testing the questionnaire, and Josette Janssen from Centerdata for testing the questionnaire on the B1 language level, adding to the questionnaire validity, and for helping out panel members who called the helpdesk whenever they needed assistance filling out the questionnaire.

**Conflicts of Interest:** The authors declare no conflict of interest as there is a mutual interest between authors and the funder in thorough scientific reporting of facts and figures about a safe sports climate. The funders had no role in the design of the study; in the collection, analyses, or interpretation of data; in the writing of the manuscript, or in the decision to publish the results.

## Notes

[1] The Dutch translation of 'once in a while' (in Dutch: een enkele keer) can mean 'once,' 'seldom,' or 'once in a while'.

[2] The LISS panel is a representative sample of Dutch individuals who participate in monthly internet surveys. The panel is based on a true probability sample of households drawn from the population register. Households that could not otherwise participate are provided with a computer and internet connection. A longitudinal survey is fielded in the panel every year, covering a large variety of domains, including health, work, education, income, housing, time use, political views, values, and personality (Scherpenzeel and Das 2010). The I&O Research panel uses the same method. More information about the LISS panel can be found at: www.lissdata.nl; about the I&Oresearchpanel at www.iopanel.nl.

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
