# Peer review of "Transgressive Behavior in Dutch Youth Sport"

_socsci, doi:10.3390/socsci11080348_

Round 1
Reviewer 1 Report
This is an excellent paper. It makes a significant contribution to the scholarship of understanding institutional settings of child sexual abuse, psychological and physical harm. The paper is expertly written. The text is clear, concise and engaging. The paper makes excellent reference to the prior study, clearly explaining changes in method. I recommend publication with no changes required.
Author Response
Thank you very much for the positive rating of our manuscript, positive remarks on its relevance and your recommendation to publish it with no changes required. However, we have included some changes to the introduction paragraph in response to review 2.
Reviewer 2 Report
This is an interesting study which principally aims to assess the prevalence of retrospectively self-reported experienced transgressive behavior in organized youth sport in the Netherlands. Additionally, the authors have investiagated for eventual differences in gender, age, disability, sport type, sport level, relationship to the offender and type of transgressive behavior. I found the subject very important, and the manuscript is presented without especial problems; therefore, I believe it is important to be published in Social Sciences after some minor modification.
Introduction
The authors are advised to proceed with a more extensive and recent literature review and include in this section some definitions, law etc., from the Methodology section. Clearly state eventual differences with Vertommen et al. 2016 (previous study).
Author Response
Thank you for your careful considerations of our manuscript, your positive remarks on its relevance and for addressing issues within the paper that needed extension or clarification. We have revised the introduction paragraph accordingly.
Please see the attachement.
